# Pediatric Septic Arthritis of the Knee Due to a Multi-Sensitive *Streptococcus pyogenes* Strain Responsive to Clindamycin—A Case Report

**DOI:** 10.3390/children8030189

**Published:** 2021-03-03

**Authors:** Giada Maria Di Pietro, Irene Maria Borzani, Sebastiano Aleo, Samantha Bosis, Paola Marchisio, Claudia Tagliabue

**Affiliations:** 1Department of Pathophysiology and Transplantation, University of Milan, 20122 Milan, Italy; paola.marchisio@unimi.it; 2Radiology Unit, Pediatric Division, Fondazione IRCCS Ca’ Granda Ospedale Maggiore Policlinico, 20122 Milan, Italy; irene.borzani@policlinico.mi.it; 3Paediatric Highly Intensive Care Unit, Fondazione IRCCS Ca’ Granda Ospedale Maggiore Policlinico, 20122 Milan, Italy; sebastiano.aleo94@gmail.com (S.A.); samantha.bosis@policlinico.mi.it (S.B.); claudia.tagliabue@policlinico.mi.it (C.T.)

**Keywords:** septic arthritis, *Streptococcus pyogenes*, clindamycin, children

## Abstract

Septic arthritis is an inflammatory process usually generated by a bacterial infection. The knee is one of the most frequently involved joints. The etiology varies depending on age, and hematogenous spread remains the primary cause in children. Herein, we report a case of a previously healthy three-year-old female who was referred to our institution for acute swelling of her right knee. After a clinical and radiological diagnosis of septic arthritis, an empirical treatment with a combination of cefotaxime and clindamycin was initiated. The isolation of a multi-sensitive *Streptococcus pyogenes* strain from the joint’s effusion prompted the discontinuation of clindamycin and the usage of cefotaxime alone. One week later, an ultrasound was executed due to worsening in the patient’s clinical conditions, and an organized corpuscular intra-articular effusion with diffuse synovial thickening was revealed. Cefotaxime was therefore replaced with clindamycin, which improved the symptoms. Despite the antibiotic sensitivity test having revealed a microorganism with sensitivity to both cephalosporin and clindamycin, clinical resistance to cefotaxime was encountered and a shift in the antimicrobial treatment was necessary to ensure a full recovery. This case study confirms that an antibiotic regimen based solely on a susceptibility test may be ineffective for such cases.

## 1. Introduction

Septic arthritis (SA) is an inflammatory process generated by a bacterial or fungal infection. The incidence of SA in children ranges from 1 to 20 cases per 100,000 children depending on the geographic region, with developing countries having the highest numbers [1,2].

SA occurs most commonly in young children, especially males. Several risk factors have been identified, such as prematurity, umbilical artery catheterization, presence of central venous catheters, and history of preceding trauma [3,4,5]. The incidence is increased by the presence of bacteremia or concomitant osteomyelitis in immunocompromised patients and those with sickle cell disease [3,4,5]. The hip and knee are the most commonly involved joints, and irreversible damage may occur if not promptly diagnosed and treated [6].

Microorganisms can enter the joint space by hematogenous spread, direct inoculation during procedures, such as arthrocentesis or intra-articular corticosteroid injection, or extension of a contiguous focus of infection due to open fractures or traumatic injuries [7]. Hematogenous infections remain the primary cause of SA in children [8].

Looking at the current literature, a recent review states that pathogens from blood and synovial fluid cultures are isolated in 34–82% of cases [4]; thus, in approximately 18–70% of cases, no organisms can be identified [9,10]. One factor contributing to culture-negative SA is *Kingella kingae* bacterial arthritis, which is difficult to culture with standard laboratory culture techniques [11].

The etiology varies depending upon age, immunization status of the patient, and geographic region. Among children between three months of age and five years old, *Staphylococcus aureus* is the most common cause of SA in children, with the methicillin-resistant *Staphylococcus aureus* (MRSA) being responsible for an increasing portion of these infections [12]. The other microorganisms responsible for SA are *Kingella kingae*, *Streptococcus pyogenes*, *Streptococcus pneumoniae,* and *Haemophilus influenzae* type B (Hib), which is especially common in areas with low Hib immunization rates [13,14].

Herein, we report the case of a patient with SA of the knee induced by a multi-sensitive *Streptococcus pyogenes*, not responsive to cefotaxime but successfully treated with clindamycin.

## 2. Case Report

A previously healthy three-year-old female was referred to our emergency room for an acute swelling of her right knee associated with limping and no history of trauma. The parents reported high fever, malaise, and pharyngodynia, which all started four days prior. Antibiotic therapy with amoxicillin/clavulanic acid was started two days before admission due to a positive rapid strep test. The girl appeared in good general conditions, with a temperature of 37.3 °C, a heart rate of 130 beats/minute, and oxygen saturation of 98%. The physical examination showed edema and hyperemia of the tonsils and pharynx, no palpable lymphadenopathies, swelling, pain, tenderness, and limited mobility of the right knee. All other joints were normal. The blood tests showed a C-reactive protein (CRP) of 21 mg/dL (normal value <0.5 mg/dL), an erythrocyte sedimentation rate (ESR) of 80 mm/h, a white blood cell count (WBC) of 9250/mmc, with 49.4% neutrophils, 43% lymphocytes, and a mild increase in liver transaminase. A blood culture and a nasopharyngeal swab tested negative.

To exclude the possible presence of fractures, an X-ray was performed of the knee, which, other than showing a mild swelling of the soft tissues in the right knee, was negative (ruling out signs of early bone involvement). A knee ultrasound detected fluid accumulation in the suprapatellar recess (Figure 1). The echocardiography and abdominal ultrasound were normal.

The patient was hospitalized in the pediatric department and an ultrasound-guided joint aspiration was performed under conscious sedation in a standardized sterile manner, yielding 8 mL of serosanguinous drainage that contained 12,000/mmc leukocytes (with prevalence of polymorphonuclear cells). This minimally invasive procedure was performed in order to establish a microbiological diagnosis (the synovial fluid was sent for WBC count and culture); at the same time, arthrocentesis decompressed the joint, helping to alleviate the child’s pain. An intravenous (IV) empiric antimicrobial treatment with cefotaxime 100 mg/kg/die and clindamycin 30 mg/kg/die was initiated. Culture performed three days later resulted positive for a *Streptococcus pyogenes* strain sensitive to all antibiotics tested (Table 1). Based on the results of the antibiotic sensitivity test and due to the clinical improvement and reduction in the CRP values (3.6 mg/dL), clindamycin was interrupted and only cefotaxime was continued.

Contrast-enhanced magnetic resonance imaging (MRI) demonstrated a small amount of fluid in the suprapatellar recess, with irregular thickening and homogeneous enhancement of the synovium. An initial phlogistic bone involvement of the distal femoral epiphysis was also noted (Figure 2).

Immunological investigation, including immunoglobulins, IgG subclasses, and lymphocyte subpopulations, resulted normal. The orthopedic evaluation ruled out surgical emergencies and the necessity of open arthrotomy.

One week after hospitalization, there was a worsening of the knee swelling and an elevation of the CRP values to 6 mg/dL. The ultrasound revealed an organized corpuscular intra-articular effusion with a diffuse synovial thickening (Figure 3). Cefotaxime was replaced with clindamycin, which improved the symptoms and normalized the CRP levels. The antibiotic treatment was continued for a total of five weeks, both intravenously and orally. The ultrasound performed one week after the antibiotic shift confirmed a gradual decrease in the joint effusion. During hospitalization, repeated orthopedic evaluations were performed, a physiotherapy treatment was initiated, and nonsteroidal anti-inflammatory drugs were administered upon requirement. The patient was discharged in good general condition and was able to walk independently.

The MRI performed one month later showed a persistence of the hyperintensity of the medial condyle of the right femur. A complete radiological resolution was observed two months after the onset of symptoms.

## 3. Discussion

Although SA usually presents with fever, weakness, immobility, or inability to bear weight on the affected joint, age can affect the way signs and symptoms are shown. Small children, for example, often exhibit unspecific features such as swelling, loss of movement in the affected joint, and excessive crying when held by parents [15]. In many cases, it is difficult to establish a differential diagnosis between SA and transient synovitis due to the similar clinical manifestations. In order to differentiate SA from transient synovitis, Kocher et al. published four parameters with a high predictive value for the diagnosis of SA of the hip: Fever >38.5 °C, ESR value >40 mm/h, WBC count >12,000/mmc, and non-weight-bearing status [16]. Caird et al., in 2006, identified a CRP level >2.0 mg/dL as a strong independent risk factor of SA and added it to Kocher’s parameters. The presence of three out of these five criteria predicts SA of the hip in 83% of cases, the positivity of four in 93% of cases, while if all five factors are observed, there is a 98% chance of diagnosing septic arthritis [17]. As these criteria were developed for the hip and not for other joints, Obey et al. published a study aimed at investigating if the sensitivity of Kocher’s criteria plus CRP could rule out the diagnosis of SA of the knee [18]. Obey et al. reported that “three or more criteria” among the five reported above, used in clinical practice for the hip, would not be sufficient to predict septic arthritis of the knee. As an early diagnosis of SA may prevent destruction of the cartilage and progression of the infection to the adjacent bone tissue leading to a permanent impairment of the knee joint, we applied the “Kocher’s criteria plus CRP level” to our patient in order to predict the rate of destructive sequelae. On admission, the child had a temperature <38.5 °C and a WBC count <12,000/mmc, but a CRP >2 mg/dL and an ESR >40 mm/h, and had difficulty walking on the affected limb. The presence of only three of the criteria suggested a 72% probability of SA [18].

Due to the increase in antibiotic-resistant bacterial strains, it is necessary to pose an accurate etiological diagnosis. The current gold standard is the identification of a microorganism in the joint fluid or in blood culture, in addition to clinical or radiological findings compatible with bacterial arthritis [19,20,21]. Regarding children with acute symptoms and elevated CRP (>2.0 mg/dL) or ESR (>20 mm/h) values, Pääkkönen created a diagnostic algorithm for SA in which arthrocentesis is used to detect any purulent aspirate and to obtain a bacteriological sample [22]. Similarly, Yagupsky et al. and Manz et al. recommended a prompt aspiration of the affected joint both to obtain a sample for bacteriological diagnosis and to achieve decompression of the articular space [23,24]. Considering that synovial fluid cultures are positive in as many as 50–60% of cases, the collected sample should also be sent for a WBC count: A count higher than 50,000/mmc in the synovial fluid increases the likelihood of SA; however, a WBC count lower than 50,000/mmc is not sufficient to rule out SA [23,24,25].

Since cultures may remain negative in as many as 30–70% of cases, children with a clinical or radiological suspicion of SA should begin an empirical antibiotic therapy [26]. Simultaneously, a blood culture should always be performed, because it is easy to obtain and repeat and may reveal the causative organism in patients with negative bone or synovial fluid cultures. Blood cultures are positive in up to 50% of cases [23,24].

Concerning imaging techniques, an X-ray should be the first imaging technique used to rule out bone involvement and other underlying conditions (fractures). Ultrasonography is useful for detecting joint effusions, synovial thickening, and soft tissue swelling, even if the presence of fluid is not specific for septic arthritis. Ultrasonography can be also used to guide diagnostic aspiration [1,19]. Other tests such as MRI and computed tomography (CT) scans should be ordered in case of diagnostic uncertainty or suspected complication [19]. An MRI provides better resolution than an X-ray or CT for the detection of joint effusion and for distinguishing between bone and soft tissue infections [24,27]. The findings seen on MRI images include joint effusion, destruction of cartilage, the presence of cellulitis in the surrounding soft tissues and osteomyelitis [27].

The treatment of SA should be started without delay after synovial fluid and blood samples have been obtained, with high-dose empirical IV antibiotics [21,22,28].

The choice of an empirical antimicrobial therapy is based on the most likely causative pathogens according to the patient’s age, immunization status, underlying disease, Gram stain of the aspirate, and regional microbiological profile, including the prevalence of MRSA in the community. Being the most common cause of SA, an antibiotic against *Staphylococcus aureus* or MRSA should be always included [19,21].

The empiric treatment for children three months or older should cover *Staphylococcus aureus* and other Gram-positive organisms. The first choice is an anti-staphylococcal penicillin or a first-, second-, or third-generation cephalosporin depending on the severity of the illness. When the MRSA prevalence for a specific region is >10–15%, then clindamycin should be added. If local clindamycin resistance rates are >10–25%, or in a setting of clindamycin inducible resistance, vancomycin should be preferred. Additional antimicrobics should be considered in children not immunized against Hib or in children between 3 and 36 months of age who could be infected by *Kingella kingae* [19,21,29].

In our case, we initially chose a combination of cefotaxime and clindamycin. After isolation of the specific pathogen and due to its susceptibility pattern, we continued with cephalosporin alone. A worsening of the symptoms and the rise in the CRP value, however, prompted us to stop cefotaxime and restart clindamycin, with benefit. As in our case, Yokoyama et al. reported a patient with SA of the hip caused by a multi-sensitive *Streptococcus pyogenes* strain, not responsive to aminobenzylpenicillina but responsive to clindamycin [30]. We hypothesized that the *Streptococcus pyogenes* isolated in our patient could be a strain producer of exotoxins that were inhibited only by clindamycin.

The IV antimicrobial regimen should be continued until clinical and laboratory improvement is seen. Sequential CRP measurement, repeated radiologic evaluations, and daily physical examinations offer useful information in monitoring recovery and the treatment’s efficacy. The duration of antibiotic therapy can range from two to seven weeks, although many studies have recommended a shorter IV regimen followed by a prolonged oral therapy [22,31,32]. In any case, the shift to an oral therapy should be considered as soon as possible when clinically appropriate [33].

The therapy’s duration depends upon the severity of illness, the pathogen, and the patient’s response. Antibiotic courses of three to four weeks in duration are usually adequate for uncomplicated bacterial arthritis. Treatment duration should be extended to six weeks if there is imaging evidence of accompanying osteomyelitis [34]. The frequency of osteomyelitis complications in children with septic arthritis is 21–42% [35].

In uncomplicated SA and in easily accessible joints, arthrocentesis may be performed with minimal morbidity and may not require general anesthesia. When abscesses, thick pus, or pus surrounding soft tissues are present, surgical drainage is indicated. However, if these are quickly diagnosed and there are no other complications, join aspiration through arthroscopy is a practical, less invasive, and effective alternative. The technique is associated with minimal soft tissue disruption, a shorter hospital stay, and improved joint space visualization are its key advantages. Open arthrotomy can be reserved for cases that do not respond to repeated aspirations [9,22,36,37,38].

## 4. Conclusions

This case study confirmed that an antibiotic regimen based only on a susceptibility test may be ineffective for such cases. The patient’s clinical condition and the laboratory tests should be the first consideration in choosing the correct antibiotic regimen.

## Figures and Tables

**Figure 1 children-08-00189-f001:**
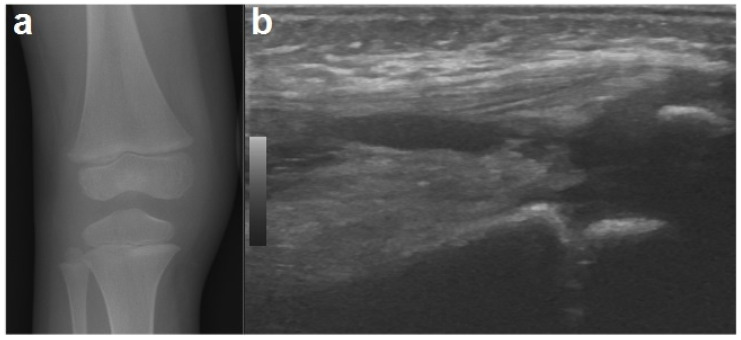
An X-ray of the right knee (**a**) showing periarticular soft tissue swelling without bone involvement. A joint ultrasound of the right knee (**b**) demonstrating a small amount of joint effusion in the suprapatellar recess.

**Figure 2 children-08-00189-f002:**
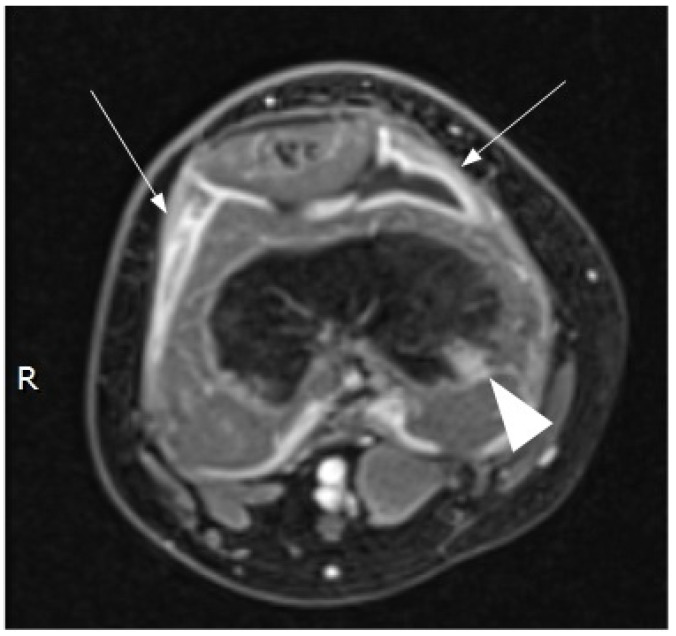
Magnetic resonance imaging (MRI) of the right knee (contrast-enhanced, axial, T1 fat-suppressed image) confirmed the presence of fluid in the suprapatellar bursa surrounded by a thickened synovial line with homogeneous contrast enhancement (arrows). Moreover, MRI detected a small bone flogistic localization in the distal epiphysis of the right femur.

**Figure 3 children-08-00189-f003:**
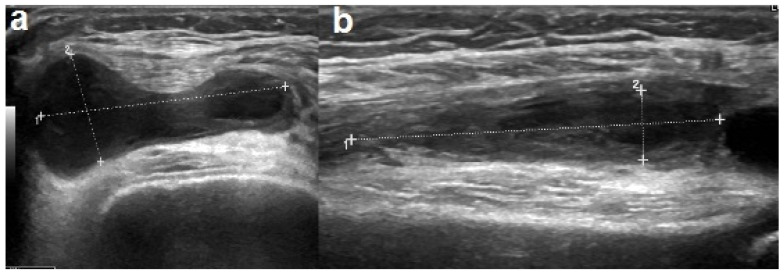
Ultrasound performed after a worsening of the right knee swelling showed increase in synovial thickening (**a**) and organized joint corpuscular effusion (**b**).

**Table 1 children-08-00189-t001:** Antibiotic susceptibility test of *Streptococcus pyogenes*.

Antimicrobial Agent	Antibiotic Susceptibility
Amoxicillin	Sensitive
Amoxiclav	Sensitive
Ampicillin	Sensitive
Azithromycin	Sensitive
Cefaclor	Sensitive
Clarithromycin	Sensitive
Clindamycin	Sensitive
Erythromycin	Sensitive
Levofloxacin	Sensitive
Penicillin	Sensitive

## Data Availability

The data presented in this study are available on request from the corresponding author. The data are not publicly available due to patient’s privacy.

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
