# Peer review of "Pediatric Septic Arthritis of the Knee Due to a Multi-Sensitive Streptococcus pyogenes Strain Responsive to Clindamycin—A Case Report"

_children, 2021, doi:10.3390/children8030189_

Round 1
Reviewer 1 Report
A Paediatric Septic Arthritis of The Knee Due to A Multi Sensitive Streptococcus Pyogenes Strain Responsive to Clindamycin
This review is based on the CARE guidelines for Case Reports published by Riley et al. (Journal of Clinical Epidemiology 89 (2017) 218e235).
I would like to thank you for the opportunity to review this interesting case. The question is relevant to the field and educational.
To my opinion, there are details of the paper that could still be improved. First, there is a lack of thoughts on why the initial treatment failed though the germ should have been sensitive to the chosen antibiotic. Secondly, the authors obviously chose not to perform a joint lavage at two critical time-points during the treatment of the young patient. This is an unusual approach and could rise questions about the quality of treatment if not justified conclusively.
Point-by-point-Review:
Title: The CARE guidelines recommend that the words “case report” or “case study” should appear in the title along with phenomenon of greatest interest (e.g., symptoms, diagnoses, tests, interventions).
Line 35: Is there a validated source for the fact “especially males…”?
Line 37: Is there a validated source for the fact “risk factors…”?
Line 72: The white cell blood count: 109/L (rather than 109/L)
Line 78: Was the joint aspiration performed in a standardized sterile manner? In an OR setting? Was there lavage of the joint performed? How many probes have been taken and how many were positive? How was containment excluded?
Line 79: “12,000/mm3” – I suggest “12.000/mmc” to maintain the same units throughout the text.
Line 82: Could you provide the antibiogram for the detected Streptococcus pyogenes?
Line 91: Based on which algorithm/checklist did the surgeons rule out the necessity to perform an arthrotomy or arthroscopic irrigation/debridement?
Line 94: Based on which thoughts did you decide not to perform a lavage of the joint at that given moment?
Line 118: Could you provide the literature reference for Obey et al.?
Line 120/121: What are the conclusions/recommended actions of Obey et al., if “three or more criteria” are not sufficient?
Line 123: “we applied the same criteria…” – The reference is not clear. Which criteria do you refer to?
Line 127: The new paragraph should begin here, rather than in line 129.
Line 136: “An x-ray should be the first imaging technique used” – Is this international consensus? What information would you expect of an X-ray at the beginning of SA? Why is ultrasonography your second choice?
Line 143/144: “SA’s treatment includes surgical drainage and lavage of the joint collecting specimens for culture…”- Could you refer this statement to your own case? Did you perform surgical drainage/lavage of the knee joint? If not, why? Why did you not perform a lavage of the knee at first presentation?
Line 153/155: Please provide the reference for this guideline.
Line 158 following: What are your thoughts on why your initial treatment with cefotaxime failed? Did you treat the right germ? Thoughts about tissue accessibility? Please highlight the strengths and limitations of the management of this case.
Line 183: Did the patient give informed consent? Please provide if requested.
Thank you very much for the opportunity to review this paper.
Author Response
We thank the reviewer for his/her valuable suggestions. The manuscript has been amended accordingly.

Reviewer 2 Report
The authors describe the case of a three-year-old girl with septic arthritis due to streptococcus pyogenes in the context of preceding pharyngitis. Empiric antibiotic treatment with clindamycin and cefotaxime was followed by monotherapy with cefotaxime in accordance with the antibiogram leading to worsening of symptoms. The switch to clindamycin resulted in full recovery of the child. The authors concluded that "clinical resistance" of antibiotics may affect recovery, that the antibiotic regimen based only on antibiogram may be ineffective "for such cases", and that "the clinical condition and the laboratory tests should be the first consideration in choosing the correct antibiotic regimen."
Major comments:
I agree with the authors and appreciate the importance of presenting a case of septic arthritis in a young child in this special issue of "Children" as diagnosis and appropriate management of septic arthritis in children remains a struggle in many emergency rooms.
I am wondering why the authors chose the broad spectrum antibiotic cefotaxime after having received the sensitivity results of the joint aspirate identifying sensitive streptococcus pyogenes instead of just using penicillin or ampicillin. This issue should be clarified due to the emergency of avoiding broad spectrum antibiotics as much as possible in the context of development of multi-resistant bacteria.
I am wondering about clinical and laboratory criteria helping in the decision making process of choosing the correct antibiotic treatment and what would be the antibiotics under consideration. I am wondering whether the authors refer to factors that may influence antibiotic efficacy such as tissue penetrance. This should be discussed as it is the main conclusion of the paper.
Further comments:
Unfortunately, the authors did not fully describe the joint aspirate; in particular, differentiation of the leukocytes and results of microscopy and gram stain are missing. The cell count was relatively low and should be discussed as important to recognize that not all infectious joints have cell counts above 50.000.
I am also wondering whether the following paragraph is misleading: "To accurately diagnose SA, the current gold standard is the identification of a microorganism in the joint fluid or blood culture in children with other clinical or radiologic findings compatible with bacterial arthritis". Isn't the joint aspiration more about identifying the particular organism to aim for targeted therapy as oppose to actually make a diagnosis of septic arthritis in general. Treatment would be initiated in any ways - even if the culture is negative.
I would like to ask to be careful about excluding osteomyelitis on day 1 or 2 of symptoms by an x-ray as described here as radiologic signs may need several days to occur.
I am not sure what the authors mean by "organized corpuscular intraarticular effusion"; this image should be added to the figures. Those results may influence the clinical course, i.e. clinical worsening, and any further treatment such as surgical interventions. The authors state that initially an orthopedic emergency was ruled out, which I assume is that orthopedics stated against the drainage of the joint, but clarification would be helpful. Also was an orthopedic consult considered with the new finding of the organized effusion?
I am wondering why the immunological studies were done as this was a first infection in a previously healthy child who was suggested to have streptococcal pharyngitis as possible portal of entrance.
Even though I appreciate that it is not the focus of this article I am wondering whether the child received any supportive therapy such as analgesics/ NSAIDs and physiotherapy.
The third figure does not have a legend.
Author Response

(The authors gave the same response as above.)

Round 2
Reviewer 1 Report
Dear authors,
I recommend to publish the paper after English editing.
Sincereley, xx
Author Response
Dear Editor,
We submitted to MDPI for English editing our manuscript.
Thank you,
Kind Regards
Di Pietro Giada